# OpenReview forum: "Transstratal Adversarial Attack: Compromising Multi-Layered Defenses in Text-to-Image Models"
_NeurIPS.cc/2025/Conference — NeurIPS 2025 spotlight_

### Official Review · Reviewer_ix2j · 2025-06-28

**Clarity:** 2
**Significance:** 2
**Originality:** 2
**Rating:** 5
**Confidence:** 4

**Summary:**

This paper explores the domain of attacking Text-to-Image (T2I) models, which are commonly used for general-purpose image generation. Specifically, it aims to generate Not-Safe-For-Work (NSFW) images. To counter such image generation, these models employ multi-layered defenses including prompt filters, concept erasers, and image filters. Prior research in this area has primarily focused on circumventing individual filters; the proposed approach in this paper attempts to bypass all three defenses simultaneously. The authors demonstrate the effectiveness of their method through experimentation.

**Questions:**

* Given the example provided in this paper, the prompt does not seem to come out of a natural language distribution. What impact would a text outlier detection model have on this framework?
* I might have missed this but how does the adaptive crossover step work? Given that we are working on the text domain, making the crossover text read in a coherent manner can be challenging.
* What does transstratal mean?

**Ethical Concerns:**

["NO or VERY MINOR ethics concerns only"]

**Final Justification:**

I am updating my scores since the authors partially addressed my two main concerns.

**Limitations:**

The authors discussed the limitations and potential negative impact of the presented work in the supplement.

**Quality:**

3

**Strengths And Weaknesses:**

# Strengths

* The motivation behind this research is clearly stated, with the authors proposing to attack all defense layers simultaneously and exploiting their overlapping vulnerabilities.
* The optimization strategy is well-explained and appears sound.
* The results show promise across various models and against different baselines.
* Additionally, the authors conducted transferability evaluations and ablation studies, which further strengthen our understanding of the proposed approach's efficacy.

# Weaknesses

* While the proposed framework is intriguing, its implementation may come at a cost. I would like to see an analysis of the computational expense involved in generating attack prompts.
* In typical adversarial attacks, robustness can be achieved by training models on adversarial examples. However, it is unclear whether this defense mechanism would be effective in this specific use case.
* Additionally, there are a few minor errors (typos) throughout the paper that should be addressed.

---

> ### Author Rebuttal · Authors · 2025-07-31
>
> Many thanks for your time and the valuable feedback on our submission.
>
> ## W1: Analysis of Computational Expense
>
> **R:** Our TAA framework involves two main computational components：(1) LLM-based candidate generation: We query three LLMs (GPT-4o, o1-mini, GPT-4.1) to generate implicit and subjective adversarial candidates, with each word generating 10 candidates. This query is usually fast and usually does not exceed 30 seconds. (2) Genetic optimization: We employ a genetic algorithm with a population size of 20 and a maximum of 20 iterations.
>
> To address computational concerns, we conducted timing experiments comparing TAA with baseline methods below, reporting the average time for one prompt optimization. While TAA takes longer than most black-box methods, it remains significantly faster than white-box approaches. Importantly, this modest computational overhead compared to black-box methods is acceptable given the improvement in attack effectiveness.
>
> | White-box | UnlearnDiff | MMA-T | PEZ   | P4D-N | P4D-K |
> |:---------:|:-----------:|:-----:|:-----:|:-----:|:-----:|
> | Time (min)| 7.66        | 19.51 | 7.94  | 15.15 | 14.23 |
>
> | Black-box | QF-Attack | DACA | Ring-A-Bell | SneakyPrompt | PGJ  | ColJailbreak | TAA  |
> |:---------:|:---------:|:----:|:-----------:|:------------:|:----:|:------------:|:----:|
> | Time (min)| 2.34      | 0.43 | 7.21        | 0.95         | 0.15 | 0.11         | 5.34 |
>
>
> ## W2: Effectiveness of Adversarial Training
>
> **R:** For defense layers in T2I models, each layer can undergo adversarial training using samples generated by TAA. Due to the time-consuming nature of this process, we primarily focus on investigating image filters. Traditional image filters fail to recognize stylistically modified harmful content. However, a classifier specifically trained on adversarial examples can learn to identify the subtle patterns and stylistic manipulations introduced by TAA.
>
> **Dataset Construction:** We created a specialized dataset named NSFW-4000 for adversarially training an image classifier. This dataset was aggregated from three sources to provide a robust training foundation:
>
> • 1,000 TAA-generated NSFW images to expose the model to the specific adversarial style
> • 1,000 benign images from the COCO dataset [a] to prevent the model from becoming overly biased and misclassifying safe images
> • 2,000 harmful images from existing public datasets [b], as referenced in the related work on unsafe image detection.
>
> **Model Fine-tuning:** We used this NSFW-4000 dataset to fine-tune the MHSC [26] image filter. The fine-tuning process is followed by their official size [c]. This creates an adversarially trained version of the classifier, which we denote as MHSC-ft.
>
> **Experimental Setup:** The experimental setup to evaluate this defense mechanism was consistent with the main evaluation in the paper. Notably, we use the fine-tuned MHSC-ft to serve as the image filter.
>
> **Evaluation Results:** The results of the adversarial training evaluation are presented below. As the table shows, the adversarial training approach demonstrates a notable improvement in defense. The fine-tuned MHSC-ft filter learned the stylistic patterns present in the subjective NSFW images and was able to block a significant portion of them. However, this defense is not a complete solution. TAA can still generate novel adversarial prompts with style variations that were not fully covered in the NSFW-4000 training set. This highlights the core challenge: adversarial training is effective against known patterns, but a generative attack like TAA can produce a near-infinite variety of styles, making it difficult for any single adversarially trained filter to achieve comprehensive protection. We would like to propose effective defense mechanisms as future work.
>
> |                     | ASC@1+1 | ASC@2+2 | ASC@4+4 |   ASR    |
> |:-------------------:|:-------:|:-------:|:-------:|:--------:|
> | TAA                 |   190   |   186   |   112   |  85.60%  |
> | Adversarial Training|   133   |   96    |   67    |  51.90%  |
>
> [a] Lin T Y, Maire M, Belongie S, et al. Microsoft coco: Common objects in context. ECCV. 2014.
> [b] A. Kim, "Nsfw image dataset," https://github.com/alex000kim/nsfw_data_scraper, 2022.
> [c] https://github.com/YitingQu/unsafe-diffusion
>
>
> ## W3: Error and Typo Correction
>
> **R:** We will carefully proofread the manuscript and address these errors and typos throughout the paper in the revised manuscript.
>
>
> ## Q1: Text Outlier Detection
>
> **R:** To evaluate the naturalness of text sequences, we use GPT-2 to calculate the perplexity (ppl) of prompt datasets. Perplexity can measure the naturalness and fluency of a text sequence, allowing us to assess the degree to which a given sentence deviates from normal language patterns.
>
> **Evaluation Results:** Beyond adversarial datasets, we include two normal T2I datasets: Midjourney-v6 [a] and DiffusionDB [b]. We randomly selected prompts from these datasets with the same number as nsfw-190. We provide the minimum, maximum, and average perplexity of each dataset below. Our analysis demonstrates that text outlier detection cannot reliably identify adversarial prompts because:
>
> • **High Degree of Text Outliers ≠ Adversarial Prompts:** We observe that DiffusionDB exhibits a larger outlier degree, yet these are not adversarial prompts. This occurs because T2I prompts often contain discrete representations for image generation, making them inherently unnatural from a linguistic perspective. For example, a typical prompt from DiffusionDB is "james graham ballard, highrise, sustainability, octane render, highly detailed," which consists of very discrete, technical terms.
>
> • **Low Degree of Text Outliers ≠ Normal Prompts:** We find that DACA achieves a smaller outlier degree, despite generating adversarial prompts. This is because DACA creates adversarial prompts entirely through LLMs, resulting in generated adversarial prompts that are more natural and fluent in language structure.
>
> Due to the inherent characteristics of T2I prompts and the sophisticated concealment techniques of existing attacks, text outlier detection models cannot efficiently and accurately distinguish adversarial prompts from normal prompts.
>
> |              | ppl-min | ppl-max | ppl-avg |
> |:------------:|:-------:|:-------:|:-------:|
> | midjourney-v6| 25      | 4975    | 424     |
> | diffusiondb  | 74      | 14212   | 717     |
> | UnlearnDiff  | 14      | 42002   | 2043    |
> | MMA-T        | 87      | 49275   | 7464    |
> | PEZ          | 55      | 170652  | 14294   |
> | P4D-N        | 152     | 28171   | 5575    |
> | P4D-K        | 261     | 29532   | 3282    |
> | QF-Attack    | 28      | 29647   | 2188    |
> | DACA         | 12      | 141     | 32      |
> | Ring-A-Bell  | 2327    | 106104  | 15168   |
> | SneakyPrompt | 18      | 84789   | 5247    |
> | PGJ          | 14      | 74026   | 2527    |
> | ColJailBreak | 21      | 1001443 | 24289   |
> | TAA          | 152     | 2981    | 627     |
>
> [a] https://huggingface.co/datasets/CortexLM/midjourney-v6
> [b] Wang Z J, Montoya E, Munechika D, et al. DiffusionDB: A Large-scale Prompt Gallery Dataset for Text-to-Image Generative Models. ACL. 2023.
>
>
> ## Q2: Clarification of Adaptive Crossover Step
>
> **R:** **Adaptive Crossover:** The crossover process works by having a child inherit candidate selection patterns from its parents based on their relative fitness scores. The mathematical formulation T_child = Crossover(T_parent1, T_parent2, f(T_parent1)/(f(T_parent1)+f(T_parent2))) means that the child is more likely to inherit successful candidate choices from the higher-fitness parent. For each word position requiring substitution, the crossover determines which parent's candidate selection strategy to follow, rather than attempting to splice text segments together.
>
> **Coherence of TAA:** Our crossover mechanism preserves semantic meaning without causing excessive modifications to the original prompt structure. Each substitution maintains semantic intent while fitting naturally into the sentence. For example, "nude" → "birthday suit" preserves the core meaning while using more implicit language. As demonstrated in Q1, our generated prompts maintain reasonable naturalness compared to existing T2I datasets and adversarial datasets.
>
>
> ## Q3: Explanation of "transstratal"
>
> **R:** "Transstratal" means cross-layer. In the context of our work, it refers to our attack methodology that simultaneously targets and exploits vulnerabilities across multiple defense layers rather than focusing on individual layers in isolation.

---

> > ### Comment · Reviewer_ix2j · 2025-08-09
> > **Response after Rebuttal**
> >
> > I thank the authors for responding to my comments. In particular, they have partially addressed my main two concerns with respect to computational cost and adversarial training. Thus I have decided to update my scores

---

> > > ### Author Response · Authors · 2025-08-09
> > >
> > > We thank the reviewer for their positive feedback and for updating their score. We are glad that our revisions were helpful in addressing their concerns.

---

### Official Review · Reviewer_6dXe · 2025-06-30

**Clarity:** 3
**Significance:** 2
**Originality:** 2
**Rating:** 4
**Confidence:** 2

**Summary:**

The paper proposes a black-box framework that exposes the critical vulnerability of overlapping weaknesses in multi-layered defenses of modern T2I models.  It generates transstratal adversarial prompts through implicit and subjective candidate generation and genetic optimization, achieving sota for average attack success rate.

**Questions:**

Why Image Filter used for IBC is single for each setting, not multiply one like Prompt Filter?
Please see more in weakness,

**Ethical Concerns:**

["NO or VERY MINOR ethics concerns only"]

**Final Justification:**

The authors address my most concerns so I raised my rating. But I am not very confident and will adjust if other reviewers have other opinions.

**Limitations:**

yes

**Quality:**

3

**Strengths And Weaknesses:**

Pros:
(1)The proposed TAA framework demonstrates higher effectiveness in bypassing multi-layered defenses of various T2I models compared to existing methods.
(2)The evaluation is comprehensive, including 14 diverse T2I models, 7 safety modules, transferability studies and so on.

Cons:
(1) The Image Harmful Score (IHS) relies on “multiple evaluators,” also IHC is evaluated by NSFW image evaluator, but the paper lacks details on evaluator.
(2) The details of some tables and Figure is missing or confusing.
a. The second row of Figure 3 without appropriate subscript.
b. There is a number 1180 in Table 4.
c. For Table 6, not explain why only these three metrics, although I think maybe you can only get these metrics.
d. For Figure 4 and 5, the captions is mixed in one line.
e. For Table 4 & 7. and Figure 5. which T2I model do you use? For Table 6, which open-source model do you use for transferability?
(3) Since it’s updated only for few steps, I think most hyper-parameters setting would be important, but some parameters in Attack setting are not presented in Figure 5.

---

> ### Author Rebuttal · Authors · 2025-07-31
>
> Many thanks for your time and the valuable feedback on our submission.
>
> ## W1: Clarification of Image Evaluators
>
> **R:** We provide detailed clarifications below regarding our choice of evaluators, the rationale behind our experimental settings, and additional results with multiple image filters:
>
> • **NSFW Image Evaluator Details:** In this paper, we have six NSFW image evaluators, including two NSFW image classifiers (NudeNet [22], NSFW-Image [7]), three CLIP-based classifiers (Q16 [33], Q16-FT [26], and MHSC [26]), and a cross-modal detection method (Safety-Checker [29]). In the main experiment (Table 2), we use the safety-checker as the image filter, and use the remaining five evaluators to calculate IHS to optimize the adversarial prompts. IHC is calculated with the same five evaluators.
>
> • **Explanation of IBC Settings:** We follow existing works (such as MMA [38], SneakyPrompt [39]), which typically utilize only one prompt filter and one image filter to attack. However, in our evaluation, we find that existing prompt filters are too weak, making it much easier to bypass one prompt filter. However, one image filter is not easy to bypass. Thus, in our experimental settings, we use multiple prompt filters and one image filter.
>
> • **Experiments with More Image Filters:** In Table 4, we have evaluated each one NSFW image evaluator as the image filter. To evaluate the TAA performance on multiple image filters (remaning image evaluators to calculate IHC), we have included corresponding results below. Our experimental results demonstrate that TAA remains highly effective against multiple image filter combinations. The defense effectiveness of multiple image evaluators is predominantly determined by the strongest individual evaluator in the ensemble, rather than achieving significantly enhanced protection through combination. The attack success rates against multiple image filters closely mirror the performance against the strongest individual filter.
>
> |                           | PBC@1 | PBC@2 | PBC@4 | IHC@1 | IHC@2 | IHC@4 | IBC | ASC@1+1 | ASC@2+2 | ASC@4+4 |   ASR    |
> |:--------------------------:|:-----:|:-----:|:-----:|:-----:|:-----:|:-----:|:---:|:-------:|:-------:|:-------:|:--------:|
> |    Safety-Checker + Q16    |  190  |  190  |  189  |  190  |  190  |  147  | 185 |   185   |   185   |   142   |  89.8%  |
> |      Q16-FT + MHSC         |  190  |  190  |  190  |  190  |  181  |  33   | 190 |   190   |   181   |   33    |  70.9%  |
> |   NudeNet + NSFW-Image     |  190  |  190  |  190  |  190  |  190  |  138  | 190 |   190   |   190   |   138   |  90.9%  |
> | Safety-Checker + Q16 + Q16-FT |  190  |  190  |  190  |  190  |  180  |  38   | 172 |   172   |   172   |   38    |  67.0%  |
> | MHSC + NudeNet + NSFW-Image |  190  |  190  |  190  |  190  |  190  |  76   | 148 |   148   |   148   |   76    |  65.3%  |
>
> ### W2: Table and Figure Correction
>
> **R:** We will correct these tables and figures as:
>
> • a)-d): We will modify these tables and figures in the revised manuscript.
> • e): For Table 4, the T2I model is SD-v1.5; For Table 7, Figure 5 and Table 6, the T2I model is SD-v1.4.
>
> ### W3: Impact of TAA Hyper-parameters
>
> **R:** The hyperparameters in TAA mainly fall into two categories: genetic algorithm parameters and LLM capability.
>
> • **Genetic Algorithm Hyperparameters:** In Figure 5, we present the analysis of key genetic algorithm hyperparameters, including candidate list size, population size, and iteration number. As shown in the results, when these evolutionary hyperparameters vary within a reasonable range, they do not significantly impact TAA's performance once they reach certain thresholds.
>
> • **LLM Capability as the Dominant Factor:** However, the primary factor influencing TAA performance is the capability of the underlying LLM used for candidate generation. In Table 14, we evaluate TAA performance under different LLM settings, which demonstrates that more powerful LLMs contribute significantly to TAA optimization. Specifically, stronger LLMs (like GPT-4.1 with 72.8% ASR) substantially outperform weaker ones (like GPT-4o with 50.4% ASR) because they generate more qualified implicit and subjective adversarial candidates that better fulfill the transstratal requirements.

---

> > ### Comment · Reviewer_6dXe · 2025-08-05
> >
> > The authors address my most concerns so I raised my rating. But I will make my final after discussing with other reviewers.

---

> > > ### Author Response · Authors · 2025-08-05
> > >
> > > Thank you for your positive feedback on our revisions. We understand the process and are ready to address any further points that may arise after your discussion with the other reviewers.

---

### Official Review · Reviewer_s2UX · 2025-07-01

**Clarity:** 2
**Significance:** 3
**Originality:** 2
**Rating:** 4
**Confidence:** 3

**Summary:**

The paper introduces Transstratal Adversarial Attack (TAA), a novel black-box framework that systematically bypasses multi-layered safety defenses in text-to-image (T2I) models. While prior adversarial attacks target individual layers, they fail against the combined system. TAA overcomes this by generating transstratal adversarial prompts (TAPs) using large language models (LLMs) and optimizing them via a genetic algorithm. These prompts use implicit NSFW expressions to bypass text filters and stylistic modifications to evade image filters, while still producing harmful content perceptible to humans.

**Questions:**

1. The proposed genetic optimization approach outperforms gradient-based methods in the experiments. Could the authors provide a deeper analysis or intuition as to why this is the case?

2. Is the genetic algorithm more effective at bypassing text-based defenses or image-based filters? Additionally, is there any observed interaction or dependency between bypassing at the text level and bypassing at the image level? For instance, does successfully bypassing the text filter increase the likelihood of bypassing the image filter, or are they largely independent?

**Ethical Concerns:**

["NO or VERY MINOR ethics concerns only"]

**Final Justification:**

The authors have addressed all my concerns, so I’m inclined to recommend accepting this work.

**Limitations:**

TAA requires iterative black-box optimization, meaning it necessitates multiple queries to the target system.

**Quality:**

3

**Strengths And Weaknesses:**

- Strengths
1. The paper introduces the first black-box adversarial attack that effectively bypasses multi-layered defenses in real-world text-to-image (T2I) systems, and provides sufficient experimental validation to demonstrate the feasibility of the attack.
2. The paper is clearly written and well-organized.

- Weaknesses
1. The notation is somewhat inconsistent or unclear in several parts of the paper:

    a. What is the Crossover function in line 214？

    b. What is p_{ind} in Equation(7), is it should be T_{ind}?

    c. What does S_{per} mean in Section 5.4? Is it S_{sub}?

2. There are also existing works that optimize prompts for language models in black-box settings—for example, [1]. Could the authors clarify how their optimization process differs from these prior approaches?


    [1] Guo et al., Connecting Large Language Models with Evolutionary Algorithms Yields Powerful Prompt Optimizers,


3. The explanation of IHC  and IBC is somewhat confusing. If a generated image is flagged as NSFW by the image evaluator, doesn't that imply the image fails to bypass the filter? In that case, shouldn’t the relationship between IHC and IBC satisfy IHC=1−IBC?

---

> ### Author Rebuttal · Authors · 2025-07-31
>
> Many thanks for your time and the valuable feedback on our submission.
>
> ## W1: Explanation and Correction of Notations
>
> **R:** We acknowledge these issues and provide the following clarifications and corrections:
>
> • a) The crossover function implements adaptive crossover where a child individual inherits genes from parents with a probability proportional to their fitness. For each candidate location, the child has a higher chance of inheriting from the higher-fitness parent and a low chance from the lower-fitness parent.
> • b) The p_{ind} in Equation(7) is T_{ind}
> • c) The S_{per} in Section 5.4 is S_{sub}
>
> ## W2: Clarification of Existing Works
>
> **R:** We clarify the fundamental distinctions between our TAA method and general prompt optimization approaches, as well as highlight our contributions relative to existing adversarial attack methods:
>
> • **Differences from EvoPrompt (Guo et al.):** Our TAA method differs fundamentally from general prompt optimization approaches like EvoPrompt [a] in several key aspects:
> 1) **Application Domain and Purpose:** While EvoPrompt focuses on optimizing prompts for improving legitimate task performance on general NLP tasks, our TAA is specifically designed for adversarial attacks against multi-layered safety defenses in T2I models. EvoPrompt aims to enhance model capabilities, whereas TAA aims to exploit vulnerabilities in safety mechanisms.
> 2) **Optimization Objectives and Specialized Generation:** EvoPrompt optimizes for conventional task performance metrics, while our TAA optimizes for a fundamentally different dual objective: maximizing bypass rate and image harmful score simultaneously. We employ domain-specific adversarial candidate generation strategies tailored for multi-layered T2I defense systems.
>
> • **Differences from Existing Adversarial Attacks:** Based on Section 2, existing adversarial attacks have critical limitations:
>
> 1) **Single-Layer vs. Multi-Layered Targeting:** Existing methods focus on isolated defenses (DACA [5] on prompt filters, UnlearnDiff [41] on concept erasers, SneakyPrompt [39] on image filters). Our TAA simultaneously targets all three layers through coordinated prompt generation.
> 2) **Holistic Strategy:** Unlike gradient-based (MMA [38]) or simple substitution methods (ColJailBreak [20]), we uniquely combine LLM-guided semantic understanding with evolutionary optimization for systematic cross-layer vulnerability exploitation.
>
> ## W3: Explanation of IBC and IHC
>
> **R:** IBC and IHC use different image evaluators, and there is no direct inverse relationship between them.
>
> • **IBC:** measures whether an image successfully bypasses the defense image filter deployed in the T2I system. This is the actual defense mechanism (e.g., Safety-Checker [29], Q16 [33], MHSC[26]) that the T2I system uses to block NSFW content.
>
> • **IHC:** measures whether the generated image is classified as NSFW by external evaluation classifiers. These are separate evaluators used only for assessment purposes, not part of the defense system.
>
> A successful attack should bypass the defense filter (high IBC) while still generating harmful content (high IHC). Therefore, IHC ≠ 1-IBC, as they measure different aspects using different evaluators. This design enables a more comprehensive evaluation of the attack's effectiveness.
>
> ## Q1: Analysis of Genetic and Gradient-based Methods
>
> **R:** The superior performance of our genetic optimization approach over gradient-based methods stems from fundamental architectural and mathematical challenges inherent in attacking multi-layered defense systems. Our experimental results (Table 2) demonstrate that while gradient-based methods can achieve reasonable performance against isolated defense components, they experience dramatic performance degradation when confronting the complete multi-layered pipeline. This performance gap can be attributed to：
>
> • **Multi-layered Defense Complexity:** Gradient-based methods are inherently limited to single-layer optimization and fail catastrophically when facing multi-layered defenses. As shown in Table 2, methods like UnlearnDiff, MMA_T, and PEZ achieve strong performance on individual defense layers but struggle with the complete pipeline because gradients computed for bypassing one layer often conflict with requirements for bypassing other layers. Additionally, backpropagating through multiple sequential defense layers (prompt filters → concept erasers → image filters) causes gradient signals to either vanish or explode, making optimization infeasible.
>
> • **Discrete-Continuous Domain Mismatch:** A critical limitation of gradient-based approaches is handling the discrete nature of text prompts. While T2I models operate on continuous embeddings, the input space consists of discrete tokens, making it difficult to accurately backpropagate from the continuous embedding space to the discrete prompt modifications. Our genetic algorithm naturally handles discrete optimization through candidate selection and mutation operations, treating the system as a black box that requires only fitness evaluation rather than gradient computation.
>
> ## Q2: Attack Relationship Among Different Defense Layers
>
> **R:** We provide a more comprehensive analysis of the genetic algorithm's effectiveness and the interdependencies between defense layers:
>
> • **Attack Effectiveness on Text-based Defense vs. Image-based Defense:** The genetic algorithm in TAA is not designed to be more effective at bypassing individual defense layers, but rather to optimize across all layers simultaneously. As shown in Table 2, the LLM-based method, DACA, achieves the best results to bypass prompt filters.
>
> • **Attack Relationship Among different Defense Layers:** To evaluate the interaction or dependency between bypassing the text-level defense and image-level defense, we select results in the optimization process from TAA that meet IHC@2 (This confirms this is a harmful case). We can observe that as the probability of bypassing more prompt filters increases, the probability of bypassing image filters also increases. This is because a lower PBC value indicates explicit harmfulness, thus making the generated images apparently harmful to image filters. Conversely, a higher PBC value can hide the harmfulness in the background, which does not trigger image filters.
>
> | IHC@2 |      IBC=0      |      IBC=1      |
> |:-----:|:---------------:|:---------------:|
> | PBC@0 | 183 (77.5%)     | 53 (22.5%)      |
> | PBC@1 | 326 (79.3%)     | 85 (20.7%)      |
> | PBC@2 | 191 (74.9%)     | 64 (25.1%)      |
> | PBC@3 | 84 (63.6%)      | 48 (36.4%)      |
> | PBC@4 | 53 (53.5%)      | 46 (46.5%)      |

---

> > ### Comment · Reviewer_s2UX · 2025-08-05
> >
> > Thank you for your response, which addresses my concerns. I will maintain my score.

---

> > > ### Author Response · Authors · 2025-08-05
> > >
> > > We sincerely appreciate your time and valuable feedback. Thank you for confirming that our revisions have addressed your concerns.

---

### Official Review · Reviewer_kPdc · 2025-07-03

**Clarity:** 3
**Significance:** 2
**Originality:** 2
**Rating:** 4
**Confidence:** 3

**Summary:**

This paper looks at a practical security threat, assuming that multi-layer protection systems are deployed instead of single-layer protection. Transstratal Adversarial Attack (TAA) is proposed, taking different defense strategies into account. In particular, TAA includes two key steps: 1. Leveraging LLMs to generate implicit and subjective adversarial candidates to substitute for prompts. 2. Including adversarial genetic optimization that provides a black-box search. Extensive experiments and ablations are conducted. Especially, the evaluation of real-world commercial T2I services is provided.

**Questions:**

- Provide discussions on the transfeability of TAA
- Discuss tailored defenses

**Ethical Concerns:**

["NO or VERY MINOR ethics concerns only"]

**Final Justification:**

This paper examines a practical threat, focusing on real systems. Extensive experiments are conducted with a carefully designed approach.

**Limitations:**

Well discussed

**Quality:**

3

**Strengths And Weaknesses:**

Strengths:

This paper looks at a practical security threat assuming that multi-layer protection systems are deployed instead of single-layer protection. It is an important step to provide a real-world security evaluation. The attack is well  designed with clear adversarial objectives. More importatntly, the proposed attack is evalueted on real-world commercial T2I systems. Related work and research background are also well-organized. Extensive experiments are conducted to evaluate the performance of TAA.

Weaknesses:

- The details of TAA transferability could be further clarified, in particular, how different defense layers are bypassed. Regarding the results in Table 6, why is TAA performing better than the other baselines? An ablation study similar to the eraser analysis can be done to figure out the exact cause for the successful attack. I would anticipate that the insights from the ablation study would be of interest to the community. In addition, would it be possible or necessary to evaluate the performance of all attacks in Table 1?

- Discussion on tailored defenses that take TAA properties into account would be interesting. For example, the implicit adversarial and subjective candidates' principles are straightforward. A defender could potentially deploy similar strategies. I would anticipate that introducing TAA-related defense may influence the normal performance. Discussions on this would be helpful.

---

> ### Author Rebuttal · Authors · 2025-07-31
>
> Many thanks for your time and the valuable feedback on our submission.
>
> ## W1 & Q1: Analysis of Transferability and Extended Results of Table 6
>
> **R:** To address your concerns about TAA's transferability mechanisms and superior performance, we have conducted comprehensive ablation studies and extended our evaluation to include all baseline methods from Table 1:
>
> • **Analysis of Transferability:** We provide an ablation study of different method components below. Specifically, we use adversarial prompts from our ablation experiments (Table 7) to evaluate transferability to commercial models. Our ablation study on commercial T2I models reveals the critical components for TAA's transferability:
>
> |                | DALL·E-2 | DALL·E-2 | DALL·E-2 | DALL·E-3 | DALL·E-3 | DALL·E-3 | Midjourney-6.1 | Midjourney-6.1 | Midjourney-6.1 | Midjourney-7 | Midjourney-7 | Midjourney-7 |
> |:--------------:|:--------:|:--------:|:--------:|:--------:|:--------:|:--------:|:--------------:|:--------------:|:--------------:|:------------:|:------------:|:------------:|
> |                |   IBC    |  ASC@1   |   ASR    |   IBC    |  ASC@1   |   ASR    |      IBC       |     ASC@1      |      ASR       |     IBC      |    ASC@1     |     ASR      |
> | w/o genetic    |    30    |    17    |   8.9%   |    21    |    4     |  2.1%   |       28       |       11       |      5.8%      |      28      |      6       |     3.2%     |
> | w/o $S_{imp}$    |    8     |    3     |   1.8%   |    5     |    2     |   1.1%   |       15       |       4        |      2.1%      |      15      |      2       |     1.1%     |
> | w/o $S_{sub}$    |    78    |    45    |  23.7%   |    69    |    31    |  16.3%   |      102       |       72       |     37.9%      |     102      |      58      |    30.5%     |
> | w/o $F_{Select}$ |    80    |    42    |  22.1%   |    60    |    27    |  14.2%   |       97       |       69       |     36.3%      |      97      |      62      |    32.6%     |
> | TAA            |    95    |    56    |  29.5%  |    81    |    36    |  18.9%  |      114       |       90       |     47.4%     |     114      |      72      |    37.9%     |
>
>
> Random candidate sampling (w/o genetic) causes significant performance drops, showing that successful attacks require careful selection. The implicit candidate set ($S_{imp}$) is most critical, as its removal leads to poor performance since explicit NSFW terms are easily caught by filters. The subjective candidate set ($S_{sub}$) provides crucial stylistic augmentation that helps images evade detection while remaining inappropriate. Failure-driven selection has moderate impact, primarily improving targeted model performance. TAA's superior performance stems from combining implicit substitution and subjective manipulation, creating prompts that are semantically harmful yet syntactically benign across defense layers.
>
> • **Extended Results of Table 6:** We evaluated all methods from Table 1 for transferability, with results consistent with Table 6. White-box methods show almost no transferability because they rely on model-specific gradients and generate syntactically irregular prompts that don't generalize. Black-box methods generate more natural prompts that bypass prompt filters and return more images, but these images have low harmfulness rates since these approaches focus on filter evasion without considering image-level harmfulness. In contrast, TAA achieves superior performance by simultaneously optimizing for both defense evasion and image harmfulness through our dual-objective design, enabling better transferability and more harmful image generation across different models.
>
> |              | DALL·E-2 | DALL·E-2 | DALL·E-2 | DALL·E-3 | DALL·E-3 | DALL·E-3 | Midjourney-6.1 | Midjourney-6.1 | Midjourney-6.1 | Midjourney-7 | Midjourney-7 | Midjourney-7 |
> |:------------:|:--------:|:--------:|:--------:|:--------:|:--------:|:--------:|:--------------:|:--------------:|:--------------:|:------------:|:------------:|:------------:|
> |              |   IBC    |  ASC@1   |   ASR    |   IBC    |  ASC@1   |   ASR    |      IBC       |     ASC@1      |      ASR       |     IBC      |    ASC@1     |     ASR      |
> | UnlearnDiff  |    3     |    1     |   0.2%   |    27    |    7     |   1.4%   |       0        |       0        |      0.0%      |      0       |      0       |     0.0%     |
> | MMA-T        |    17    |    2     |   1.1%   |    65    |    15    |   7.9%   |       11       |       7        |      3.7%      |      11      |      8       |     4.2%     |
> | PEZ          |    28    |    4     |   2.1%   |    69    |    14    |   7.4%   |       0        |       0        |      0.0%      |      0       |      0       |     0.0%     |
> | P4D-N        |    13    |    2     |   1.1%   |    38    |    18    |   9.5%   |       0        |       0        |      0.0%      |      0       |      0       |     0.0%     |
> | P4D-K        |    1     |    0     |   0.0%   |    16    |    6     |   3.2%   |       0        |       0        |      0.0%      |      0       |      0       |     0.0%     |
> | QF-Attack    |    3     |    0     |   0.0%   |    14    |    3     |   0.6%   |       0        |       0        |      0.0%      |      0       |      0       |     0.0%     |
> | DACA         |   109    |    16    |   8.4%   |   124    |    8     |   4.2%   |      110       |       24       |     12.6%      |     110      |      19      |    10.0%     |
> | Ring-A-Bell  |    0     |    0     |   0.0%   |    7     |    1     |   0.2%   |       0        |       0        |      0.0%      |      0       |      0       |     0.0%     |
> | SneakyPrompt |    50    |    2     |   0.4%   |    66    |    10    |   3.0%   |       42       |       4        |      1.2%      |      42      |      3       |     0.7%     |
> | PGJ          |    77    |    13    |   6.8%   |    88    |    16    |   8.4%   |       0        |       0        |      0.0%      |      0       |      0       |     0.0%     |
> | CoJailBreak  |    78    |    18    |   9.5%   |    64    |    12    |   6.3%   |       71       |       34       |     17.9%      |      71      |      23      |    12.1%     |
> | TAA          |    95    |    56    |  29.5%   |    81    |    36    |  18.9%   |      114       |       90       |     47.4%      |     114      |      72      |    37.9%     |
>
> ## W2 & Q2: Defense Enhancement Strategies for TAA
>
> **R:** The effectiveness of TAA stems from two primary vulnerabilities it exploits: 1) Implicit NSFW Prompts: These prompts bypass filters by using metaphorical or subtle language while retaining their harmful generative capability. 2) Subjective NSFW Images: These images are crafted to evade automated classifiers but are still perceived as harmful by humans.
>
> Based on these attack designs, we investigated tailored defense strategies:
>
> • **LLM-Based Processing for Implicit Prompts:** As you noted, a defender can use a similar strategy to the attacker. TAA uses LLMs to generate implicit, harmful prompts. As a countermeasure, we deployed an LLM processor (specifically, GPT-4o) as an additional defense layer. This processor is tasked with analyzing and sanitizing prompts by identifying and removing the subtle, metaphorical NSFW language that TAA creates.
>
> • **Adversarial Training for Subjective NSFW Images:** To counter subjective NSFW image generation, we employed adversarial training. Standard image filters often fail as they are trained on objectively explicit content. By training on TAA-generated adversarial examples, we can teach classifiers to recognize TAA's subtle patterns and style modifications. We constructed a NSFW-4000 dataset with 1,000 TAA-generated NSFW images, 1,000 benign COCO images, and 2,000 harmful images from existing datasets. We then fine-tuned MHSC [26] with NSFW-4000 according to the official implementation, generating MHSC-ft.
>
> **Experimental Setup:** We use the same setup as in Table 2, with the following modifications: we incorporate a GPT-4o-based LLM processor after the prompt layers for LLM Processing, and employ MHSC-ft as the image filter for adversarial training. For normal task performance evaluation, we collected prompts from the Midjourney-v6 [d] dataset matching the size of nsfw_190. We measure prompt-image similarity using ClipScore and image fidelity using FID-30k scores on Midjourney-v6. For defense performance evaluation, we apply the attack metrics from Section 5.1 on the nsfw_190 dataset.
>
> **Evaluation Results:** We provide preliminary defense results for LLM processing and adversarial training. LLM processing shows limited effectiveness against TAA-generated prompts. While it handles simple transformations like converting "a birthday suit woman" to "a woman," it struggles with TAA's complex, context-dependent prompts. Adversarial training demonstrates stronger performance by recognizing stylistic patterns in subjective NSFW images, but remains insufficient since TAA can generate infinite style variations that exceed the coverage capacity of trained filters. Effective defense mechanisms remain as future work.
>
> |                      | ClipScore (↑) | FID-30k (↓) | ASC@1+1 | ASC@2+2 | ASC@4+4 |    ASR   |
> |:--------------------:|:-------------:|:-----------:|:-------:|:-------:|:-------:|:--------:|
> | Base                 |     0.788     |    15.2     |    -    |    -    |    -    |     -    |
> | TAA                  |       -       |      -      |   190   |   186   |   112   |  85.6%  |
> | LLM Processing       |     0.782     |    16.3     |   164   |   141   |   97    |  70.5%  |
> | Adversarial Training |     0.788     |    15.2     |   133   |   96    |   67    |  51.9%  |
>
> [a] Lin T Y, Maire M, Belongie S, et al. Microsoft coco: Common objects in context. ECCV. 2014.
> [b] A. Kim, "Nsfw image dataset," https://github.com/alex000kim/nsfw_data_scraper, 2022.
> [c] https://github.com/YitingQu/unsafe-diffusion
> [d] https://huggingface.co/datasets/CortexLM/midjourney-v6

---

### Official Review · Reviewer_CKn6 · 2025-07-04

**Clarity:** 2
**Significance:** 2
**Originality:** 2
**Rating:** 4
**Confidence:** 4

**Summary:**

The paper proposes an adversarial attack technique called Transstratal Adversarial Attack (TAA), designed to bypass the multi-layered defenses commonly employed in modern Text-to-Image (T2I) systems, which are responsible for preventing the generation of Not Safe For Work (NSFW) content. These systems typically employ multiple defense layers such as prompt filters, concept erasers, and image filters.

**Questions:**

1. The paper could provide a more detailed section on ethical concerns and guidelines for responsible use of this research.
2. The paper evaluates TAA against a wide range of T2I models (14 models), but what about models that use different architectures or non-diffusion-based methods? How would TAA fare against newer models that are not part of this evaluation?
3. Does the proposed attack maintain similar success rates when applied to newly developed or proprietary T2I models that have not been encountered during training?

**Ethical Concerns:**

["Major Concern: Data privacy, copyright, and consent", "Major Concern: Deception and harassment"]

**Final Justification:**

Overall, the authors have effectively addressed most of my concerns, particularly those related to the novelty, defense layer analysis, adversarial image perceptibility, and TAA's performance across different models. Based on their responses, I would suggest giving the authors a chance to revise the paper with a more robust ethical discussion and additional improvements in defense strategies. The paper has strong contributions but would benefit from further clarification in some areas. Hence, I raised my rating.

**Limitations:**

1. The paper demonstrates the vulnerabilities in these systems, and it raises significant ethical concerns regarding the potential for misuse.
2. The framework does not fully explore alternatives or propose concrete solutions for improving internal defenses.
3. The NSFW-related dataset used for generating adversarial prompts (nsfw_190) includes a limited set of examples. It may not cover the full spectrum of NSFW content and therefore, the success of the attack could be restricted to the types of prompts present in this dataset.
4. Extending the evaluation to include other harmful outputs beyond NSFW content, such as misleading or harmful political content, would make the research more holistic.

**Quality:**

3

**Strengths And Weaknesses:**

Strength:
TAA targets the entire multi-layer defense pipeline of T2I systems, including prompt filters, concept erasers, and image filters, a significant step forward in adversarial research for these models. Previous attacks often targeted only one layer.

Weaknesses:
1. The paper considers multiple layers for attack. It is a combination of existing works.
2. The paper emphasizes the weaknesses in multi-layered defenses but doesn't delve into potential solutions to these weaknesses.
3. While the attack success rate (ASR) is thoroughly discussed, the visual quality of the generated images is not evaluated. How does the quality of adversarial images compare to legitimate content in terms of perceptibility and harmfulness?
4. Although the attack targets three primary defense layers—prompt filters, concept erasers, and image filters—the paper does not provide enough details on how these layers are interdependent and complementary. A more in-depth analysis of these interactions and their combined effect on security would help better understand the weaknesses of these defense mechanisms.

---

> ### Author Rebuttal · Authors · 2025-07-31
>
> Many thanks for your time and the valuable feedback on our submission.
>
> ## W1: Paper Novelty and Contribution
>
> **R:** We would like to clarify the significant novel problems and contributions of our paper:
>
> • **Novel Problem (Multi-Layered Defense Vulnerability):** Existing attacks fail against multi-layered defenses due to contradictory requirements across layers. Prior works target only 1-2 layers, making TAA the first to systematically address all three layers in a black-box setting.
>
> • **Novel Technical Contributions:** Our method introduces transstratal adversarial candidate generation that solves contradictory optimization across defense layers through LLM-guided implicit candidates, subjective candidates appearing harmful to humans but benign to classifiers, and failure-driven genetic optimization for multi-objective prompt optimization.
>
> ## W2 & L2: Defense Enhancement Strategies for TAA
>
> **R:** TAA exploits two critical vulnerabilities: implicit NSFW prompts bypassing filters while maintaining harmfulness, and subjective NSFW images evading automated detection while remaining harmful to humans. We propose direct and adaptive defense strategies:
>
> • **Direct Defense (Comprehensive Concept Removal):** The most straightforward defense would be removing NSFW concepts from T2I models. However, this is computationally infeasible due to vast semantic space and substantially degrades general image generation quality, as many concepts are intertwined with benign functionalities.
>
> • **Adaptive Defense Mechanisms:**
> 1) **LLM Processing for Implicit NSFW Prompts:** Since TAA leverages LLMs to generate implicit NSFW terms, we employ a symmetric defense, using LLMs to detect these implicit expressions. We implemented an LLM processor (GPT-4o in the experiment) as an additional layer after traditional prompt filters.
> 2) **Adversarial Training for Subjective NSFW Images:** Traditional image filters fail on stylistically modified harmful content. We fine-tuned MHSC [26] on our NSFW-4000 dataset (1,000 TAA-generated NSFW images, 1,000 benign COCO images, 2,000 harmful images from existing datasets [a]) to recognize TAA's patterns, creating MHSC-ft.
>
> **Experimental Setup:** We use the same setup as Table 2 with GPT-4o-based LLM processor and MHSC-ft as image filter. We evaluate normal tasks using Midjourney-v6 prompts [b] with ClipScore and FID-30k, and defense using attack metrics from Section 5.1 on nsfw_190.
>
> **Evaluation Results:** LLM processing handles simple implicit prompts but struggles with TAA's complex, context-dependent prompts, showing limited effectiveness. Adversarial training performs better by learning stylistic patterns but remains insufficient since TAA can generate infinite style variations exceeding trained filter coverage. Effective defense mechanisms remain future work.
>
> |                      | ClipScore (↑) | FID-30k (↓) | ASC@1+1 | ASC@2+2 | ASC@4+4 |    ASR   |
> |:--------------------:|:-------------:|:-----------:|:-------:|:-------:|:-------:|:--------:|
> | Base                 |     0.788     |    15.2     |    -    |    -    |    -    |     -    |
> | TAA                  |       -       |      -      |   190   |   186   |   112   |  85.6%  |
> | LLM Processing       |     0.782     |    16.3     |   164   |   141   |   97    |  70.5%  |
> | Adversarial Training |     0.788     |    15.2     |   133   |   96    |   67    |  51.9%  |
>
>
> [a] A. Kim, "Nsfw image dataset," https://github.com/alex000kim/nsfw_data_scraper, 2022.
> [b] https://huggingface.co/datasets/CortexLM/midjourney-v6
>
> ## W3: Perceptibility Evaluation of Adversarial Images
>
> **R:** To address the visual quality of generated images, we have evaluated prompt-image consistency using ClipScore in Table 10. For a comprehensive assessment of visual quality, we follow metrics from [39] and [13]:
>
> • **FID:** Evaluates image semantic similarity between adversarial images and two ground-truth datasets: (1) Target: 1,900 images generated by nsfw-190 prompts with ten seeds, and (2) Real: existing NSFW dataset with 4,260 real NSFW images [a].
>
> • **Semantic Consistency (SC):** Uses BLIP to evaluate the semantic alignment of generated images with their prompts.
>
> Black-box methods (including ours) achieve better prompt-image similarity by using natural language substitutions rather than optimization techniques producing unnatural prompts. Our method achieves comparable visual quality while maintaining prompt naturalness, representing an effective trade-off between attack effectiveness and perceptibility.
>
> | | ClipScore (↑) | FID (Target ↓) | FID (Real ↓) | SC (↑) |
> | :---: | :---: | :---: | :---: | :---: |
> | UnlearnDiff | 0.688 | 183.5 | 177.2 | 0.379 |
> | MMA-T | 0.692 | 207.4 | 209.9 | 0.348 |
> | PEZ | 0.698 | **157.2** | 174.9 | 0.381 |
> | P4D-N | 0.643 | 161.0 | **165.9** | 0.380 |
> | P4D-K | 0.662 | 200.9 | 215.4 | 0.386 |
> | QF-Attack | 0.677 | 235.8 | 224.4 | 0.291 |
> | DACA | 0.711 | 241.1 | 238.4 | 0.36 |
> | Ring-A-Bell | 0.711 | 217.3 | 228.9 | 0.324 |
> | SneakyPrompt | 0.705 | 235.3 | 232.1 | 0.333 |
> | PGJ | 0.733 | 231.5 | 231.5 | 0.375 |
> | ColJailBreak | **0.738** | 208.9 | 205.4 | 0.383 |
> | TAA (ours) | 0.727 | 175.8 | 181.3 | **0.387** |
>
> [a] A. Kim, "Nsfw image dataset," https://github.com/alex000kim/nsfw_data_scraper, 2022.
>
> ## W4: Defense Layer Analysis
>
> **R:** To address this concern, we conducted a comprehensive analysis of defense layer interactions using data collected from our TAA optimization process. We collected 62,460 substituted prompts generated during the TAA optimization process across all experiments in Table 2.
>
> **Strength of Defense Layers:** Our analysis reveals the defense strength hierarchy: Concept Erasers (CE) > Image Filters (IF) > Prompt Filters (PF). This exists because: (1) PF target explicit keywords and struggle with implicit expressions; (2) CE operate in latent space, making them harder to bypass; (3) IF are deceived by adversarial candidates maintaining semantic content while avoiding detection.
>
> | Not Bypass PF | Not Bypass CE | Not Bypass IF |
> | :---: | :---: | :---: |
> | 13,729 | 45,043 | 29,899 |
>
>
> **Independence and Correlation Among Defense Layers:** We analyzed bypass patterns across different defense layer combinations, revealing key interdependency patterns:
>
> • **Complementary Defense of CE and Outer Defenses:** Adversarial prompts bypassing outer defenses face significantly increased difficulty bypassing concept erasers, but this asymmetry doesn't work in reverse. This occurs because concept erasers detect NSFW semantics in latent space that outer defenses using explicit pattern matching cannot capture
>
> • **Strong Correlation between IF and PF:** High correlation exists between prompt and image filter bypass outcomes. This stems from both layers' shared sensitivity to explicit NSFW content, creating overlapping detection capabilities that provide redundant rather than complementary protection.
>
>
> |  | Bypass PF | Bypass CE | Bypass IF |
> |:-------------:|:--------------------:|:---------------------:|:-------------------:|
> | Bypass PF (48,731) | - | 11,884 | 27,206 |
> | Bypass CE (17,057) | 11,884 | - | 4,264 |
> | Bypass IF (32,561) | 27,206 | 4,264 | - |
>
>
>
> |  | Not Bypass PF | Not Bypass CE | Not Bypass IF |
> |:-------------:|:------------------------:|:-------------------------:|:-----------------------:|
> | Bypass PF (48,731) | - | 36,847 | 21,525 |
> | Bypass CE (17,057) | 5,173 | - | 12,793 |
> | Bypass IF (32,561) | 5,355 | 28,297 | - |
>
>
> |  | Bypass PF | Bypass CE | Bypass IF |
> |:-------------:|:--------------------:|:---------------------:|:-------------------:|
> | Not Bypass PF (13,729) | - | 5,173 | 5,355 |
> | Not Bypass CE (45,403) | 36,847 | - | 28,297 |
> | Not Bypass IF (29,899) | 21,525 | 12,793 | - |
>
>
> ## Q1 & L1: Detailed Section on Ethical concerns and Responsible use
>
> **R:** We addressed these concerns in Appendices F-H and will elevate them in the revision. Our work serves as a defensive tool exposing vulnerabilities and providing the first holistic robustness benchmark. We will explicitly acknowledge the dual-use nature and malicious exploitation potential.
>
> ## Q2 & Q3: TAA Performance on Unseen Models
>
> **R:** We added Janus-Pro [a], an auto-regressive T2I model, with results below. Our evaluation covers diverse architectures:
>
> • Diffusion models: SD series
>
> • Flow Matching models: FLUX.1-dev and FLUX.1-schnell
> • Auto-regressive models: Janus-Pro
>
> | Model | Architecture | Release Time | ASC@1+1 | ASC@2+2 | ASC@4+4 | ASR |
> |:-----:|:------------:|:------------:|:-------:|:-------:|:-------:|:---:|
> | Janus-Pro [a] | Auto-regressive | 25'01 | 190 | 190 | 96 | 83.5% |
>
> TAA demonstrates strong performance across these architectural paradigms, achieving consistent attack success rates despite fundamental differences in generation mechanisms.
>
>
>
> [a] Chen X, Wu Z, Liu X, et al. Janus-pro: Unified multimodal understanding and generation with data and model scaling. arXiv, 2025.
>
> ## L3: Evaluation of More NSFW Categories
>
> **R:** We focused on sexual content in nsfw_190 as it's a primary safety concern creating a challenging testbed for multi-layered defense evaluation and ensures fair comparison with prior works [39, 13].
>
> Following your suggestion, we extended evaluation to violence, harassment, and self-harm categories. Our content-agnostic attack generalizes well to these domains using 200 GPT-4o generated prompts per category, following [13]'s methodology. The results of this extended evaluation are presented below:
>
> |  | PBC@2 | IHC@4 | ASC@2+4 | ASR |
> |:--------:|:-----:|:-----:|:-------:|:---:|
> | Violence | 188 | 52 | 52 | 71.3% |
> | Harassment | 200 | 76 | 76 | 79.3% |
> | Self-harm | 200 | 56 | 56 | 74.3% |
>
> These results demonstrate TAA's high efficacy across diverse harmful content categories, confirming our method exploits systemic vulnerabilities in multi-layered defenses rather than being restricted to specific domains.

---

### Decision · Program_Chairs · 2025-09-17

**Decision:**

Accept (spotlight)

**Comment:**

The paper proposes a black-box attack, Transstratal Adversarial Attack (TAA), that targets multi-layered T2I defenses by combining LLM-generated implicit and subjective prompt candidates with failure-driven genetic optimization. The evaluation results show that TAA achieves an average ASR of 85.6% and outperforms SOTA attacks across 14 models and 17 safety modules.

The work is well-scoped around realistic, end-to-end pipelines and is evaluated broadly, including commercial systems, with clear evidence of transferability advantages over strong baselines.

Reviewers asked for clearer analysis of transferability and tailored defenses and the authors have addressed these during the rebuttal with ablations and extended results. They have also experimented with an LLM-based prompt processor and adaptively trained image filter to probe defenses.

Given the realistic threat model, comprehensive evidence, and that reviewer concerns were largely addressed, I recommend Accept. Please incorporate all reviewers’ comments into the final version and, per the ethics reviewers, add an Ethics Statement.